# SARS-CoV-2 T Cell Immunity Responses following Natural Infection and Vaccination

**DOI:** 10.3390/vaccines11071186

**Published:** 2023-06-30

**Authors:** Vassiliki C. Pitiriga, Myrto Papamentzelopoulou, Kanella E. Konstantinakou, Kalliopi Theodoridou, Irene V. Vasileiou, Athanasios Tsakris

**Affiliations:** 1Department of Microbiology, Medical School, National and Kapodistrian University of Athens, 75 Mikras Asias Street, 11527 Athens, Greece; lmktheo@yahoo.com (K.T.); atsakris@med.uoa.gr (A.T.); 2Molecular Biology Unit, 1st Department of Obstetrics and Gynecology, National and Kapodistrian University of Athens, 75 Mikras Asias Street, 11527 Athens, Greece; mpntua@yahoo.gr; 3Bioiatriki Healthcare Group, Kifisias 132 and Papada Street, 11526 Athens, Greece; nkonstantinakou@bioiatriki.gr (K.E.K.); irenebasileiou@gmail.com (I.V.V.)

**Keywords:** cellular immunity, T cell immunity, SARS-CoV-2, COVID-19, coronavirus, vaccination, natural infection, IFN-γ, ELISpot, IGRA

## Abstract

(1) Background: SARS-CoV-2 T cell immunity is rapidly activated following SARS-CoV-2 infection and vaccination and is crucial for controlling infection progression and severity. The aim of the present study was to compare the levels of T cell responses to SARS-CoV-2 between cohorts of subjects with hybrid immunity (convalescent and vaccinated), vaccinated naïve (non-exposed) and convalescent unvaccinated subjects. (2) Methods: We performed a retrospective descriptive analysis of data collected from the medical records of adult individuals who were consecutively examined at a large, private Medical Center of Attica from September 2021 to September 2022 in order to be examined on their own initiative for SARS-CoV-2 T cell immunity response. They were divided into three groups: Group A: SARS-CoV-2 convalescent and vaccinated subjects; Group B: SARS-CoV-2 naïve vaccinated subjects; Group C: SARS-CoV-2 convalescent unvaccinated subjects. The SARS-CoV-2 T cell response was estimated against spike (S) and nucleocapsid (N) structural proteins by performing the methodology T-SPOT.COVID test. (3) Results: A total of 530 subjects were retrospectively included in the study, 252 females (47.5%) and 278 (52.5%) males ranging from 13 to 92 years old (mean 55.68 ± 17.0 years). Among them, 66 (12.5%) were included in Group A, 284 (53.6%) in Group B and 180 (34.0%) in Group C. Among the three groups, a reaction against S antigen was reported in 58/66 (87.8%) of Group A, 175/284 (61.6%) of Group B and 146/180 (81.1%) of Group C (chi-square, *p* < 0.001). Reaction against N antigen was present in 49/66 (74.2%) of Group A and in 140/180 (77.7%) of Group C (chi-square, *p* = 0.841). The median SFC count for S antigen was 24 (range from 0–218) in Group A, 12 (range from 0–275) in Group B and 18 (range from 0–160) in Group C (Kruskal–Wallis test, *p* < 0.001; pairwise comparisons: groups A–B, *p* < 0.001; groups A–C, *p* = 0.147; groups B–C, *p* < 0.001). The median SFCs count for N antigen was 13 (range 0–82) for Group A and 18 (range 0–168) for Group C (Kruskal–Wallis test, *p* = 0.27 for A–C groups). (4) Conclusions: Our findings suggest that natural cellular immunity, either alone or combined with vaccination, confers stronger and more durable protection compared to vaccine-induced cellular immunity.

## 1. Introduction

Cellular immunity is known to play a vital role in recognizing and controlling viral pathogens. Concerning severe acute respiratory syndrome coronavirus 2 (SARS-CoV-2), coordinated T cell-mediated immunity is revealed to be associated with effective viral clearance and less disease severity [1,2]. Several studies have revealed that T cells can provide protection against reinfection [3] with persistent immunologic memory following SARS-CoV-2 exposure [4] or vaccination [5]. Indeed, prior SARS-CoV-2 infection offers protection of around 87% at 6 months [6] with an unchanged profile up to at least 10 months [7].

Moreover, although a decrease in antibody titers has been commonly observed, recent studies support the prolonged protective role of T cells compared to humoral immunity [8,9]. Interestingly, in a cohort of convalescent healthcare workers, unlike the gradually decreasing antibody titers, cellular immunity levels maintained stable throughout the 9-month study period [10]. Other studies report that virus-specific T cells maintained an estimated half-life of 200 days [11]. Despite the great amount of available research data, the durability of SARS-CoV-2 cellular immunity following the coronavirus disease of 2019 (COVID-19), reinfection and vaccination still arouses great research interest.

Considering mRNA vaccines, a profound adaptive humoral and poly-specific cellular immune response against epitopes is known to be elicited [12]. Remarkably, vaccination following SARS-CoV-2 infection was recently correlated with a remarkable enhancement of both humoral and cellular immune responses [13]. More specifically, a significant increase in both cellular and humoral responses against ten SARS-CoV-2 variants was documented after vaccination of SARS-CoV-2 convalescent subjects when compared with vaccination of SARS-CoV-2 naïve subjects. Although antibody titers wane after vaccination, memory poly-specific T cell and B cell subsets are induced, providing more robust protection against severe, critical and fatal infection or reinfection [14]. However, the breadth and duration of immunologic memory require further investigation.

The durability of immune protection following SARS-CoV-2 infection and/or vaccination wanes over time and in response to novel variants of concern (VOCs) [15]. In contrast, a large amount of T cell epitopes are conserved between VOCs, facilitating improved disease control compared to the humoral response, which appears to be more easily evaded by spike protein changes [16]. Notably, T cell-mediated immunity remains quite unaffected by the antigenic variation in the viral spike glycoprotein compared to humoral responses [17].

Although cellular responses remain more difficult to study compared to humoral ones, the implementation of advanced laboratory technologies facilitates T cell response assessment, thus, providing thorough T cell immunity characterization. Such technologies are broadly divided into molecular and cellular assays. Among those, enzyme-linked immunosorbent spot assays (ELISpot), based on the detection of cytokine secretion for a single cell, have been proven particularly useful [18].

Considering that new variants will continuously emerge along with the decreased efficacy of the humoral responses, a comprehensive understanding of T cell responses to SARS-CoV-2 is crucial for improving public health strategies, while more cellular immunity-associated studies are required to establish their protective role. We conducted this retrospective study in order to compare the levels of T cell responses to SARS-CoV-2 infection between three cohorts of the population: (a) subjects with hybrid immunity (convalescent and vaccinated), (b) vaccinated naïve (non-exposed) and (c) convalescent unvaccinated.

## 2. Materials and Methods

### 2.1. Study Design

We performed a retrospective descriptive analysis of data collected from the medical records of adults who had consecutively proceeded to ‘BIOIATRIKI” Healthcare Center, a large Medical Center in the region of Attica, Greece, from September 2021 to September 2022 in order to be examined for SARS-CoV-2 T cell response upon their own initiative, as part of a SARS-CoV-2 immunity screening. Additional data in terms of clinical symptoms and medical history were obtained by structured questionnaires that were routinely filled by all individuals at the time of examination. No precise information was obtained regarding the exact type of vaccine administered; however, m-RNA vaccines were largely available to the majority of the Greek population during that period. No records of clinical data, such as chest X-rays and hospitalization records, were obtained for infected patients.

Based on the available data, we divided participants into 3 cohorts: Group A: both SARS-CoV-2 convalescent and vaccinated subjects; Group B: SARS-CoV-2 naïve subjects who were vaccinated; Group C: SARS-CoV-2 convalescent and unvaccinated subjects.

The study was approved by the institutional review board (date of approval: 29 June 2021, 6th Annual Meeting).

### 2.2. Study Groups and Clinical Definitions

(i)Group A: SARS-CoV-2 Convalescent/vaccinated group

Participants were selected among vaccinated COVID-19 patients based on a previous positive SARS-CoV-2 PCR either alone or combined with positive rapid antigen test and/or self-test result. The date of the first positive laboratory result was considered the date of diagnosis of COVID-19 infection. In an attempt to reduce the unavoidable interference of confounding factors, subjects with COVID-19 reinfections confirmed by PCR, either alone or combined with positive rapid antigen test and/or self-test result, were excluded from the study. Moreover, we included in the study only individuals who had completed their initial vaccination series and had not received any booster doses until the examination time. The date of the initial vaccination completion (date of 1st or 2nd dose administration, depending on the type of vaccine) was considered as the date of vaccination. The time interval before the T cell response measurement was estimated, taking into account the first immunological response to SARS-CoV-2, either due to infection or vaccination. No adequate information was available about the exact date of the second event of immunological response.

(ii)Group B: SARS-CoV-2 naïve (non-exposed)/vaccinated group

Participants were selected from subjects with low risk of prior SARS-CoV-2 infection and no prior diagnosis of COVID-19 infection who had been fully vaccinated. Requirements for enrollment included no current or prior signs or symptoms of COVID-19, no known contact with a confirmed SARS-CoV-2-infected individual and no diagnosis with SARS-CoV-2 prior to vaccination based on PCR results, rapid tests, self-tests and specific IgG IgM antibodies measurements. We included subjects who had completed only their initial vaccination series and had not received any booster doses until the examination time. The date of the initial vaccination completion (date of 1st or 2nd dose administration, depending on the type of vaccine) was considered as the date of vaccination.

(iii)Group C: SARS-CoV-2 convalescent/unvaccinated group

Participants were selected from unvaccinated asymptomatic and symptomatic patients with a previous positive SARS-CoV-2 PCR result prior to testing. The date of the first positive PCR result was considered the date of diagnosis of SARS-CoV-2 infection. In cases where PCR was carried out as a confirmation test after a positive self-test or rapid test, the date of the first positive laboratory test was considered as the date of SARS-CoV-2 diagnosis. Individuals with confirmed COVID-19 reinfections were excluded from the study.

### 2.3. Laboratory Diagnosis

Enzyme-linked immunosorbent spot (ELISpot) Assay for IFN-g T Cell Response Detection: The T cell response to SARS-CoV-2 infection was estimated by performing the IGRA methodology T-SPOT^®^.COVID (Oxford Immunotec, Oxfordshire, UK). The T-SPOT.COVID test is a standardized ELISpot-based technique for detection of T cell immune response to SARS-CoV-2 in whole blood. The test uses the established T-SPOT Technology with an antigen mix based on SARS-CoV-2 structural proteins, spike (S) and nucleocapsid (N).

Blood samples were drawn into lithium heparin tubes where the T-cell Xtend reagent (Oxford Immunotec) was added. Peripheral blood mononuclear cells (PBMCs) were then separated from a whole blood sample, washed and then counted before being added to the test. An estimated number of 250,000 cells/well were plated into 4 wells of a 96-well plate.

The two antigen peptide pools were added to the two antigen wells, the T cell mitogen phytohemagglutinin was added to the positive control well, and cell culture media alone to the negative control well. The wells were washed after a period of 16–20 h of incubation and were added a conjugated secondary antibody with the capacity to bind to any IFN-γ captured on the membrane. In the process, the wells were washed to remove the unbound IFN-γ, and a substrate was added to produce the characteristic dark spots of insoluble products indicating areas of IFN-γ presence. The spot-forming cells (SFCs) were manually counted by microscopy by expert and experienced technologists. Results were reported separately for N and S antigens. They were expressed as ‘invalid’ if the negative control had more than 10 SFCs or the positive control had fewer than 20 SFCs when the antigen wells were non-reactive. The test cut-off was predefined at 6 SFCs for each antigen. A borderline zone of +/−1 SFCs was introduced to account for potentially elevated test variability around the cut-off [19]. Accordingly, results were reported as S and/or N reactive’ when the SFCs in the S or/and N antigen well minus the negative control were ≥8, S and N non-reactive when the SFCs in the respective antigen wells minus the negative control were ≤4, and S or/and N borderline’ when the SFCs in the respective antigen well minus the negative control were 5, 6 or 7.

### 2.4. Statistical Analysis

The chi-square (*χ*^2^) test was performed to evaluate differences between categorical variables, and one-way ANOVA was used to compare continuous variables. In cases of non-parametric continuous variables, the Kruskal–Wallis test was performed. Kinetics over time periods were estimated using the curve estimation. GLM Univariate analyses were performed to examine the existence of confounding factors, particularly sex, age, time after exposure and cardiovascular disease.

The statistical analysis and graphical representations were carried out using SPSS version 28 (IBM SPSS Statistics). Outcomes were considered statistically significant when *p*-value was <0.05.

## 3. Results

### 3.1. Participants’ Demographic Characteristics

Among the total individuals consecutively proceeding to our medical center for SARS-CoV-2 immunity screening, 530 were selected for the study, 252 females (47.5%) and 278 (52.5%) males ranging from 13 to 92 years old (mean 55.68 ± 17.0 years). The study participants’ medical information is presented in Table 1.

Among them, 66 (12.5%) had hybrid immunity (previously infected and vaccinated) (Group A), 284 (53.6%) had been vaccinated and had no evidence of prior exposure to the virus (Group B), and 180 (34.0%) had past COVID-19 infection and had not been vaccinated (Group C).

In Group A, the sex proportion was 29 females (43.9%) and 37 males (56.1%); in Group B, 127 females (44.7%) and 157 males (55.3%); in Group C, 96 females (53.3%) and 84 males (46.7%) (chi-square test, *p* = 0.1) The mean age in Group A was 57.3 ± 16.45 years, in Group B, 58.46 ± 16.32 years, and in Group C, 50.69 ± 17.28 (one-way ANOVA test, F = 12.33 *p* < 0.01). More specifically, a significant difference in age was observed between Group A and C (*p* < 0.001) and Group B and Group C (*p* < 0.001). No difference in age was demonstrated between Group A and Group B (*p* = 0.68).

### 3.2. Positivity Rate among Total Patients and between Groups

Of the overall T cell response results, S antigen elicited a positive response in 379/530 (71.5%) and N antigen in 189/530 (35.6%) of the samples, respectively. Borderline S results were reported in 13/530 (2.4%) and non-reactive in 138/530 (26%), while borderline N results were reported in 13/530 (2.4%) and non-reactive in 328/530 (61.8%). No participants had a borderline response against both N and S antigens. Both S and N antigens triggered positive responses in 186 (35.1%) samples. In 12 samples, the results of both two antigens were considered indeterminate due to inadequate responses in the negative or positive controls. These indeterminate samples corresponded to six unvaccinated (of Group C) and six vaccinated immunocompromised cases (of Group A) and were excluded from the study.

In addition, eight samples presented less than 20 SFCs in the positive control; however, four of them had a response against the S antigen and the other four against the N antigen; therefore, they were considered positive only for the particular antigen.

Among the three groups, a reaction against the S antigen was reported in 58/66 (87.8%) of Group A, 175/284 (61.6%) of Group B and 146/180 (81.1%) of Group C (chi-square, *p* < 0.001). Reaction against N antigen was present in 49/66 (74.2%) of Group A and in 140/180 (77.7%) of Group C (chi-square, *p* = 0.841) (Figure 1a,b). Reaction against N antigen was not detected in Group B.

### 3.3. Quantitative IFN-g Response against SARS-CoV-2 Antigens

The median SFC count for S antigen was 24 (range 0–218) in Group A, 12 (range 0–275) in Group B and 18 (range 0–160) in Group C (Kruskal–Wallis test, *p* < 0.001; pairwise comparisons: groups A–B, *p* < 0.001; groups A–C, *p* = 0.147; groups B–C, *p* < 0.001). The median SFCs count for N antigen was 13 (range 0–82) for Group A and 18 (range 0–168) for Group C (Kruskal–Wallis test, *p* = 0.27) (Figure 2). Reaction against N antigen was not detected in Group B.

### 3.4. IFN-g Response According to Days after Vaccination and after COVID-19 Diagnosis

The mean time period post-exposure/vaccination for the three groups were as follows: Group A: 175 ± 124 days (range 14–600), Group B: 136 ± 76 days (range 13–324) and Group C: 254 ± 155 (range 12–705) days (one-way ANOVA, F = 36.83 *p* < 0.001). More specifically, a statistically significant difference was revealed between groups A and B (F = 10.3, *p* = 0.002), groups A and C (F = 4.9, *p* = 0.028) and groups B and C (F = 54.8, *p* < 0.001).

Kinetics over time (in days) and curve estimation of the quantitative T-SPOT results for S and N antigens in subjects of groups B and C are presented in Figure 3 and Figure 4a,b.

### 3.5. Univariate Analysis

In order to examine the possible interaction between S and N antigens and the independent variables age, sex, cardiovascular disease and time after exposure, we performed GLM univariate analysis. The constructed model did not demonstrate any associations in a statistically significant level with any of the above variables (for S antigen: Age: F = 0.678, *p* = 0.41; sex: F = 0.198, *p* = 0.657; time after exposure F = 4.373, *p* = 0.09; cardiovascular disease: F = 0.373, *p* = 0.752. For N antigen: Age: F = 2.703, *p* = 0.101; sex: F = 0.557, *p* = 0.456; time after exposure F = 2.373, *p* = 0.136; cardiovascular disease: F = 0.453, *p* = 0.859).

## 4. Discussion

Humoral and cellular immune memory assessment against SARS-CoV-2 is critical in controlling COVID-19 progress. At present, long-term immune responses concerning both types of responses in convalescent patients and vaccinated subjects have been widely investigated [20,21].

The present retrospective descriptive study has focused on the evaluation of T cell responses to SARS-CoV-2 in three different cohorts. Our results suggest that cellular immunity induced by natural infection alone or combined with vaccination is higher compared to the vaccine-induced immunity alone. These findings are of particular significance if we take into account that the time interval from initial immunity triggering to T cell testing was longer in groups A and C compared to the time interval from vaccination to testing day in Group B. Interestingly, our study suggests that an IFN T cell response, when comparing the response between Group B and Group C, is stronger in infected patients than in those who have been vaccinated. This should be further examined by future studies, as the response has been described to be variable in vaccinated subjects since vaccines may not be equally effective against SARS-CoV-2 variants [2].

With regard to the effect of vaccination on SARS-CoV-2 cellular immunity, a recent study has documented that levels of T cells remained sustained after 3 months from vaccination, whereas the titer of anti-RBD antibodies as well as their neutralization function decreased significantly during that period [22]. Additional reports [5,23] have demonstrated that levels of memory T and B cells were stable for 6 months after vaccination in contrast to reduced antibody titers. It should be noted that current vaccines have been recently documented to elicit broadly cross-reactive cellular immunity against SARS-CoV-2 variants, including Omicron [24,25].

Concerning convalescent individuals, several studies [26,27,28,29] confirmed T cell responses—either alone or along with humoral immunity persistence up to 6–12 months after the infection. In another recent study [30], SARS-CoV-2- T cell responses were detected 12 months after initial infection, with a decline in neutralizing antibody titers between 6 and 12 months after infection, mainly in older people and critical patients. These reports concerning cellular responses post-infection are in line with our results since we revealed an approximately 8-month cellular immunity in convalescent patients.

As for convalescent individuals who had received one of the EMA-approved vaccines, recent studies [5,31,32,33] report that significantly higher levels of cellular immunity and antibody titers were detected in both convalescent and vaccinated subjects compared to those who were either vaccinated or COVID-19 convalescent. Moreover, the utility of booster vaccination was highlighted elsewhere [7], with cellular immune responses being reactivated by a one-dose vaccine in SARS-CoV-2 convalescent patients.

Our study bears certain limitations, including those of an observational retrospective study. The study groups comprised of individuals proceeding to the diagnostic center on their own initiative to be tested for cellular immunity as part of a COVID-19 immunity screening test during the pandemic. In this context, no additional data, such as the type of the administered vaccines—except that the vast majority were mRNA—and the identification of viral variants, were available for further analysis. Furthermore, in Group A, no information could be obtained about the exact time of the second immunological response, which implies the need for caution in the interpretation of results.

## 5. Conclusions

Ultimately, our results enhance the evidence of long-lasting cellular immunity in COVID-19 convalescent subjects, especially when combined with vaccination. Future studies aiming to evaluate cellular responses years after vaccination and/or infection could provide further insights into longevity and persistence of SARS-CoV-2–specific T cell-mediated immunity as well as its importance against reinfections or potential new antigen encountering.

## Figures and Tables

**Figure 1 vaccines-11-01186-f001:**
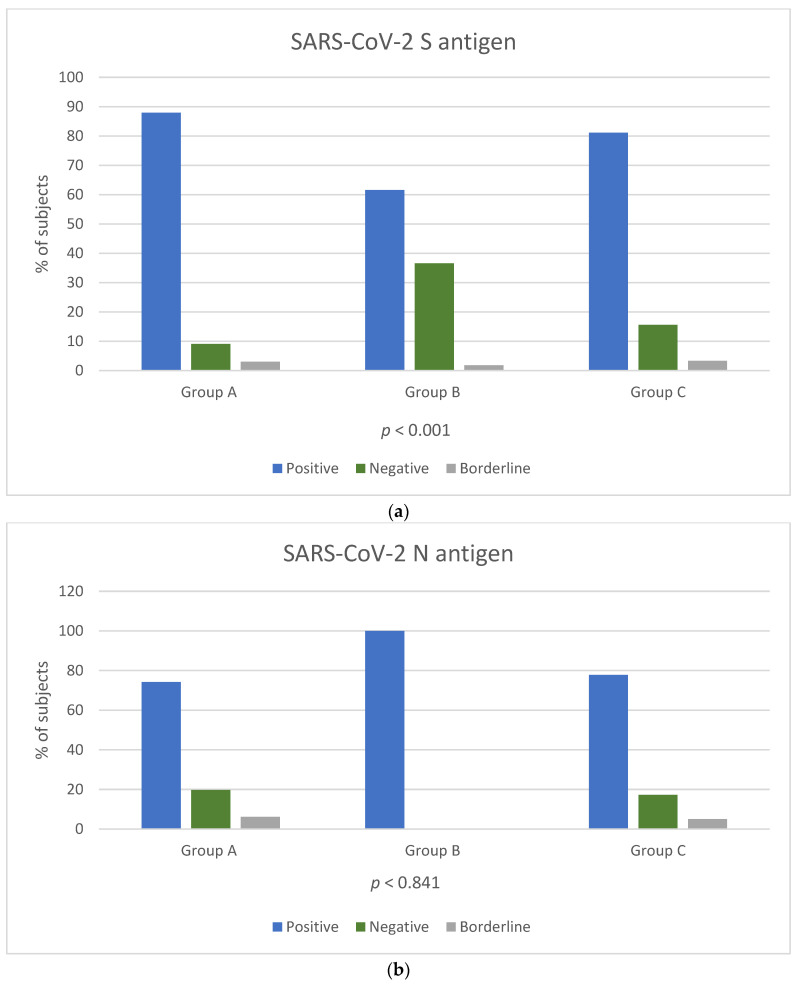
Positivity rate of T-SPOT results for cellular responses against SARS-CoV-2 S (**a**) and N (**b**) antigens among the three groups. T cell response results are expressed by three categories: positive result (blue colour bar), negative result (orange colour bar) and borderline result (grey colour bar). *p*-Values by chi-square test (*p* < 0.05 is considered to be statistically significant). *p*-Values in Figure 1b were calculated for groups A and C.

**Figure 2 vaccines-11-01186-f002:**
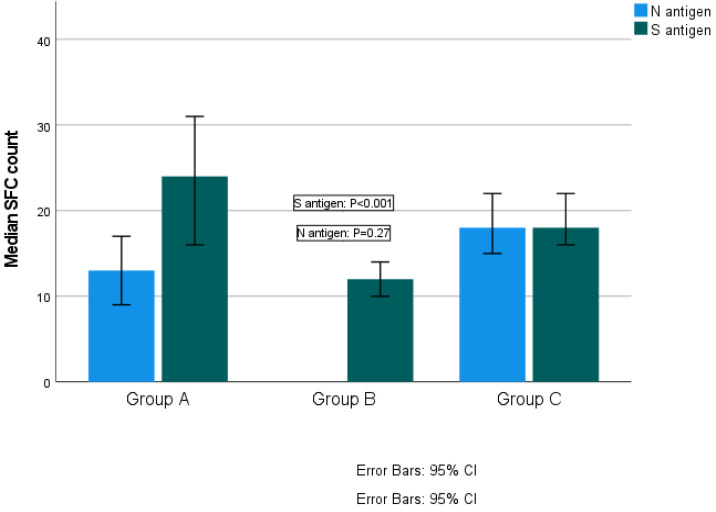
Quantitative T-SPOT results for T cell responses against SARS-CoV-2 N and S antigens among groups (*p*-values by Kruskal–Wallis test). *p*-Value < 0.05 is considered to be statistically significant.

**Figure 3 vaccines-11-01186-f003:**
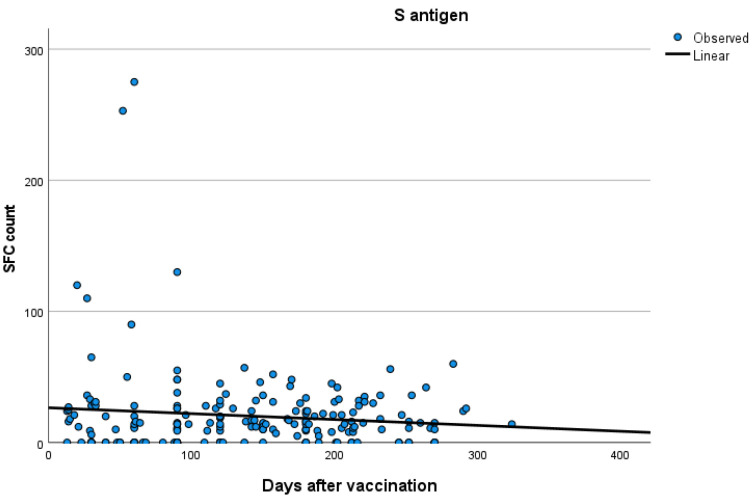
Kinetics over time (in days) and curve estimation of the quantitative T-SPOT results (dots in figures) for S antigen in subjects of Group B.

**Figure 4 vaccines-11-01186-f004:**
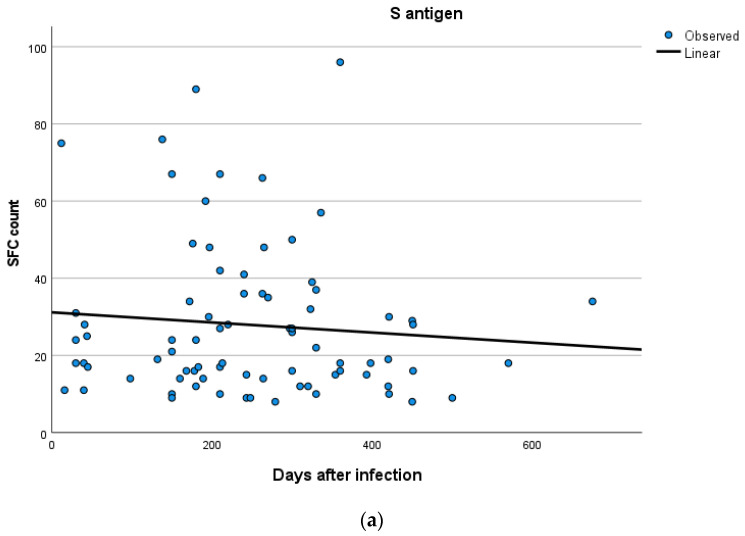
(**a**,**b**) Kinetics over time period (in days) and curve estimation of the quantitative T-SPOT results (dots in figures) for S and N antigens in subjects of Group C.

**Table 1 vaccines-11-01186-t001:** Participants’ demographic characteristics and data of clinical symptoms and comorbidities. *p*-Values by chi-square test (*p* < 0.05 is considered to be statistically significant).

Variables	Group A (*N* = 66)	Group B (*N* = 284)	Group C (*N* = 180)	*p*-Value
	*N* (%)	*N* (%)	*N* (%)	
Demographic characteristics				
Age (years ± SD)	57.3 ± 16.45	58.46 ± 16.32	50.69 ± 17.28	0.1
Sex (F/M)	29/37 (43.9/56.1)	127/157 (44.7/55.3)	96/84 (53.3/46.7)	<0.01
Time after exposure (days ± SD)	175 ± 124	136 ± 76	254 ± 155	<0.001
Clinical symptoms of infected participants				
Asymptomatic	6 (9.0)	0	35 (19.4)	
Symptomatic	60 (90.9)	0	145 (80.5)	
Shortness of breath	14 (21.2)	0	30 (16.6)	NS
Sore throat	58 (87.8)	0	138 (76.6)	NS
Fatigue	63 (95.4)	0	145 (80.5)	NS
Loss of taste/smell	35 (53.0)	0	60 (33.3)	NS
Diarrhea	15 (22.7)	0	34 (18.8)	NS
Headache	60 (90.9)	0	143 (79.4)	NS
Vomiting	10 (15.1)	0	24 (13.3)	NS
Congestion	15 (22.7)	0	43 (23.8)	NS
Fever	49 (74.2)	0	135 (75.0)	NS
Comorbidities				
Respiratory disorders	12 (18.1)	56 (19.7)	32 (17.7)	NS
Cardiovascular diseases	15(22.7)	35 (12.3)	25 (13.8)	0.04
Autoimmune disorders	18 (27.2)	47 (16.5)	37 (20.5)	NS
Central nervous system disorders	3 (4.5)	10 (3.5)	11 (6.1)	NS
Malignant neoplasia	5 (7.5)	18 (6.3)	10 (5.5)	NS
Diabetes mellitus	25 (37.8)	74 (26.0)	65 (36.1)	NS
Hypertension	39 (59.0)	130 (45.7)	100 (55.5)	NS
Lipidemia	29 (43.9)	114 (40.1)	87 (48.3)	NS
Obesity	26 (39.3)	130 (45.7)	120 (66.6)	NS
Allergies	12 (18.1)	55 (19.3)	54 (30.0)	NS
Immunosuppressive treatment	3 (4.5)	10 (3.5)	9 (5.0)	NS
Cortisol intake	1 (1.5)	4 (1.4)	5 (2.7)	NS

NS, not significant; N, number of subjects; SD, standard deviation; F/M, Female/Male.

## Data Availability

All data of this study are included in this article.

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
