# Peer review of "SARS-CoV-2 T Cell Immunity Responses following Natural Infection and Vaccination"

_vaccines, 2023, doi:10.3390/vaccines11071186_

Round 1

Reviewer 1 Report

In this paper, the authors analyzed the T-cell response specific to SARS-CoV-2 antigens in PBMC (peripheral blood mononuclear cells) derived from 3 groups of participants: Group A: SARS-CoV-2 infected and vaccinated people, group B: SARS-CoV-2 naïve individuals who were vaccinated, group C: SARS-CoV-2 infected individuals who had not been vaccinated.

They analyzed the specific T cell response by IFNg ELISPOT assay. Their results show a stronger T cells response in infected patients (group C) compared to vaccinated people (group B) suggesting that natural cellular immunity following infection confers a stronger protection compared to vaccine-induced immunity. This work is interesting and the study is well conducted. Participants characteristics, inclusion and exclusion criteria are well described…

However, I have some comments to improve the manuscript :

-        Line 108: “a confirmed SARS-CoV-2-infected individual and no diagnosis with SARS-CoV-1 prior to vaccination”: there is a mistake : SARS-CoV-1 instead of SARS-CoV-2

-        Line 176 and 177 : The same percentage of borderline results are obtained for S antigen and N antigen (2,4%). Are the same participants having a borderline response for the 2 antigens ?

-        Line 187: “Reactive N antigen was present in 49/66 (74.2%) of groups A and 140/180 of group C”. The percentage is not indicated for group C.

-        Line 236-238 : “Moreover, when comparing the response between group 2 and  group 3, our study indicates that an IFN T-cell response is stronger in infected patients than in those subjects who have been vaccinated”. Replace group 2 and 3 by group B and C.

-        Concerning the presentation of the results, I think it is necessary to add some figures illustrating the results described in part 3.2 and 3.3. Perhaps some histograms showing the positivity rate and others showing the mean SFC (spot forming cells).  

-        Can the authors explain why, in each condition, the T cell response against the N antigen seems to be weaker than the T cell response against the S antigen ?

Reviewer 2 Report

The paper by Pitiriga and colleagues is a study on the detection of specific T-cells in samples from SARS-CoV-2 infected and/or vaccinated subjects.
The introduction is sufficently explanatory, even if it only a few of the many papers on the topic have been cited and others could be added.
The methods section provide information on the study design, though many relevant details are missing (see minor points) and should be added.
The section about the IGRA assay could be more synthetic.
There's a limit of the study design: the exclusion of re-infections provides a bias towards those subjects, among the infected ones, with a better immune response, exluding
those who probably had a worst one. This bias could be responsible of all differences found between groups.
Also, info about vaccines are mssing. It is known that different vaccines have different immunological outcomes. Info about vaccines should be provided, and if different ones
were used, comparison should also be performed considering different vaccination groups. Last, information about viral variants which infected the subjects is missing.
This information, if available, could be useful for correlation with cellular immunity outcomes.
The results section lacks figures showing the data presented in paragraphs 3.3 and 3.3. Some graphic representation of those data is required.
Some more results could be shown, for example, a parallel test on antibodies (neutralization or ELISA) is highly recommendable.
The discussion is probably overenthusiastic, as shown data are very interesting but would require many confirmations, and with an improved study design, to
provide solid demonstration of the proposed assumptions. Also, needless to say, one only technique has been used, and this stronlgy limits the relevance of the results.
The paragraph, in the discussion, about the t-spot.covid assay, is unnecessarily enthusiastic and endorsing, and should be rewritten in a shorter, clearer and much more neutral form.
So, my suggestions are:
- improving m&m section by providing the missing info
- using a second assay (possibly on T-cells, at least on antibodies)
- improving subjects selection and data analysis.
- improving data presentation

Some minor points:
- line 12: ...the levels of T-cell....
- line 40-41: please rephrase in clearer form
- line 42: the referred article shows results of vaccinated subjects,not natural infections only
- line 80: the center in which patients were tested should be expicitated. Furthermore, the inclusion criteria are not clearly stated: what were these people
testing their blood for? It looks like they were tested purposely for this study. In this case, inclusion criteria must be clarified in detail.
- line 86: please rephrase in clearer form
- line 88: if the study was approved, please add details in the backmatter section (under 'Institutional Review Board Statement')
- line 89: Information about the administered vaccines should be provided
- line 90: there is not an explanation wether these subjects were infected before or after the vaccination.
Anyways, they should't be mixed in only one group, since their immunologic outcome is expected to be different.
- line 100: the second immunological trigger, not the first, should be considered for calculation of time interval. Using the first, since the interval with
the second is variable (and not provided) the extimates of the actual kinetics of IFN production over time are inaccurate.
- line 123: please define IGRA
- line 166: it is not clear why, being the study presented as a retrospective one, the authors didn't even try to obtain 3 groups of roughly the same size.
This would have eliminated many risks of stathistical biases.
- line 181: immunocompromised subjects should not have been included in the study
- line 185-187: please rephrase: antigens are not detected nor reported. Reaction against them is detected and reported
- lines 209,210, 213,216, 217: figures lack description of the y-axis
- line 225: is this study prospective, or retrospective as elsewhere stated?
- line 227: 'our study results' suggests', or something similar. Demostration is a term to be used with caution, since demostrations required a different study design
and more tests confirming each other.
- line 248-253: please rephrase. The periods are difficult to understand. If necessary add other refs
- line 265: please omit authors names and/or use correct format for reference number
- line 278: please omit authors names and/or use correct format for reference number
- line 283: 'convalescent group C'
- line 351: please reformat the authors's names

Reviewer 3 Report

In this paper Pitiriga et al. attempt to show that T cell immunity is long-lived following antigen exposure through SARS-CoV-2 infection and/or COVID-19 vaccination. They also conclude that infection, or infection and vaccination, is better than vaccination alone with regard to T cell immunity. However, the data presented is weak due to incorrect figures and inadequate statistical analyses. The overall description of the cohort and subjects used is minimal and there are some major flaws in the language. The introduction and discussion sections are very weak, and, to this reviewer, it doesn’t appear like the authors actually read any of the papers cited. This reviewer fails to see what new data this paper offers with the already broad knowledge of long-lived T cell immunity following infection (of any kind). It’s great to see that the data supports the body of literature but the lack of description and analysis of the subjects (particularly group A) makes this paper extremely weak and not supported for this journal. The authors lack basic immunological understanding by comparing functional T cell assays (ELISpot) to neutralization assays or ELISAs (i.e. comparing a functional expansion assay of cells to a protein), to argue in favor of T cell responses. The limitations addressed by the authors are major and the conclusions drawn are not very well supported by the data, primarily due to bad graphical representation and lack of statistics.

Specific comments in more detail are below:

Line 47: SARS-CoV-2 needs to be unabbreviated before being abbreviated to SARS-CoV-2. Additionally, “severe acute respiratory distress syndrome coronavirus 2 (SARS-CoV-2) virus” is not correct. Remove “virus”.

Line 49: “several studies have revealed” – for correct English.

Throughout the paper you refer to both “T-cells” and “T cells”, pick one and be consistent. T cells (without the hyphen) is more correct).

Lines 48-50: The cited papers do not show what the authors state. Both papers show longitudinal T cell responses but neither show protection from infection. Neither paper even analyzes reinfection.

Line 53: no comma after “although”.

Line 53: “is observed” is present tense English. “has been observed” is correct.

Lines 58-60: T cell immunity against severe infection and reinfection and vaccination are under investigation? There’s already a huge body of literature on these parameters in the context of T cell immunity.

Lines 61-62: “Considering mRNA vaccines… epitopes is elicited”, this sentence needs references.

Lines 62-66: “Surprisingly…SARS-CoV-2-naïve participants”, this sentence is somewhat confusing, and I would consider revising for reader clarity. The authors say “convalescent” previously and now use the term “SARS-CoV-2-recovered”, pick one and be consistent.

Lines 66-67: “Although antibody titers wane…Suggesting cellular immunity remains”, this sentence is somewhat misleading. Yes, antibody titers wane, but memory B cell subsets are also induced. So yes, memory responses from cellular immunity remains but so does humoral immunity. I could easily reforge the sentence for the counter argument: “Although cytotoxic T cell wane after vaccination, memory B cell subsets are induced, suggesting that humoral immunity remains”. Subsets from both arms of the immune system induce both acute responses which wane, and memory subsets that persist.

Line 72: “strong cross-protection” is incorrect as it implies that someone would be protected, which is completely false in the clinical setting (i.e. many that target conserved T cell epitopes get reinfected regardless). The sentence needs to be changed to say that a large amount of T cell epitopes are conserved between VOCs, thus implying that there can be improved disease control compared to the humoral response, which appears to be more easily evaded by spike protein changes.

Line 73: “quite insensitive” is not a scientific terminology. Consider changing.

Line 75: “On the other hand”, on the other hand of what? Please revise this sentence for reader clarity.

Line 79: “interferon-gamma release assay”? So, an ELISpot?

Lines 82-83: “T-cell responses to COVID-19”, you don’t make T cell responses to a disease, you make it against the pathogen. i.e. SARS-CoV-2. Additionally, COVID-19 has not yet been unabbreviated.

Line 85: “compare the levels of T-cell response”, is not correct English. Either “compare the levels of T cell responses” or “compare the levels of the T cell response”.

Line 86: what is “naturally infected”? or in other words, what is unnaturally infected? This is a nebulous term and should be removed.

Lines 84-86: To sell the study more, I would write that you compared vaccinated and naïve (not exposed), infected and vaccinated and infected and not vaccinated. I.e. you compared the T cell response between 3 groups.

Materials and methods

Line 92: “region of Attica”, please add country.

Line 93: “COVID-19 T-cell immunity response”, is not correct English, please revise. Again, it’s SARS-CoV-2 T cell responses (i.e. the pathogen not the disease).

Line 106: “nucleic acid amplification test”? So, a PCR?

Line 108: What laboratory results?

Lines 110-111: How did you identify reinfection? PCR? RAT? ELISA?

Lines 104-117: Where the participants filtered by vaccine type? (i.e. AZ, J&J, Pfizer/BioNTech, Moderna?) If yes, please state. If no, why not? It’s important to note that AZ and J&J vaccines, especially heterologous vaccination, induce far better T cells responses than homologous mRNA vaccination. There is also mention of date of 1st or 2nd vaccination, which one was it? There is a big difference in immune responses between 1st and 2nd vaccinations. Also, what is meant by “first immune triggering”?

Line 137: It’s ELISpot not “ELISPOT”, and needs to be unabbreviated.

Lines 143-145: “Fresh blood samples… manufacturers instructions”, the test requires PBMCs and states no instruction on how to isolate PBMCs from whole blood. Please remove “manufacturers instructions” and relate the sentence to the procedure below this sentence.

Line 168: can the authors please state the type of ANOVA used (i.e. parametric or non-parametric). Can the authors also please state why they used the said type of ANOVA. Assuming that a parametric ANOVA was used, did the authors check for normal distribution of the data? If not, why?

Results

Lines 175-176: “COVID-19 immunity” should be “SARS-CoV-2 immunity”.

Lines 175-177: Can the authors please add the individual groups sex and age into Table 1 (i.e. the data from lines 182-186). On line 182, “gender” is incorrect and should be “sex”. Furthermore, there is no information on the type of vaccine given, or if it is 1st dose or 2nd dose. This either needs to be in supplementary information or needs to be stated in the methods, clearly.

Line 180: “not known exposure” is incorrect English. “no evidence of prior exposure” is more scientific and more correct.

Line 181: “had past NAAT-confirmed SARS-CoV-2 infection”, the authors need to mention that this group had no follow up vaccination. Or summarize that the groups are: A) infected-vaccinated, B) vaccinated only and C) infected only. Additionally, the authors find a significant difference in cardiovascular comorbidities between groups A and C, was this adjusted for?

Line 186: A significant age difference is shown (P<0.01). Was age then adjusted for in downstream analyses? If yes, please state. If no, please explain why. There should also be three P values comparing A to B, A to C and B to C. Simply showing the significance in variance is not correct.

Lines 195-196: Please state which groups these samples belonged to.

Line 196: the authors mentioned immunocompromised cases yet have failed to mention any analysis or adjustment for immunocompromised individuals. This should be in Table 1, as this is a huge confounding variable.

Lines 197-199: PHA is an extremely good control for ELISpot and, if the samples were negative for this, should be excluded from further analysis.

Line 205: “SFC” has already been unabbreviated in the above text and “spot forming cells” in the parenthesis is not required.

Lines 205-209: Why is the mean SFC shown when a non-parametric ANOVA was used? Was the data checked for normal distribution to use the mean? If no, please used the medians and round it to the nearest spot value (i.e. you can’t have 0.66 of a spot, so make it 33 spots not 32.66). The authors only state one P value for the ANOVA, likely only showing the significance. Can the authors please show the individual P values for each group comparison (Kruskal-Wallis test) (i.e. A to B, A to C and B to C).

Lines 211-216: These time periods should be in Table 1. What is the ±? Assuming that this is the SD and the authors need to state as much. Again, it would be recommended round the values to the nearest day. There is a clear significant difference between each group, was this adjusted for in the analysis the authors conducted above? If no, this is an important confounding variable and will need to be adjusted for. The authors need to be clearer what time point was used for the analysis for group A. They have stated two different time points here, one post infection and one post vaccination. Assuming that the time point post vaccination was used, this is the one they should stick with. In addition to this, the authors need to state the mean time between infection and vaccination. Is there any way the authors can measure the T cell response in group A post infection and then post vaccination? This would be an excellent way to help remove bias when comparing infection alone to infection-vaccination.

Lines 217-224: Group A has a linear increase in S and N responses over time, which is actually against what the literature says ( https://doi.org/10.1128/jvi.00509-22 https://doi.org/10.3390/vaccines10122132 ). Is this because these responses are both after infection and vaccination? I highly doubt an acute infection (followed by vaccination), at 600 DPO, would have nearly 50 SFC. From my own data and others, it’s barely above background after a year PSO or Post-vaccination. Further, the authors report on a correlation between either antigen and time? One doesn’t correlate to time, you are observing kinetics over time and, in this case, you are looking for a linear trend, not a correlation. Lastly, you have stated “p” here and “P” previously, please be consistent with how you present p values.

Discussion

Lines 231-232: Use either “the T cell response” or “T cell responses”.

Line 232: The evaluation was done on T cell responses to SARS-CoV-2, not specifically to SARS-CoV-2 infection. I.e. group B was not infected. Also, “population” is incorrect. It was done on a cohort.

Line 234: “compared to vaccine-induced”, do the authors mean ‘vaccine-induced alone’? This conclusion is not warranted without the proper statistical information. i.e. Kruskal-Wallis test (non-parametric ANOVA) or one-way ANOVA (parametric ANOVA) showing the significant differences between each group. This conclusion is not very convincing without time-matching being taken into account.

Lines 237-242: The conclusion of increasing cellular immunity over time in group A is completely false. The “second triggering of the immune system” (I’m assuming meaning vaccination) completely misrepresents the data here. Firstly, the authors have failed to indicate at which time points they analyzed the subjects. Secondly, comparing subjects with multiple antigen exposures to subjects with a single antigen exposure over time is not warranted. The linear increase is completely misleading because there would be a decrease over time following the initial antigen exposure (as seen by the body of literature and in Groups B and C), and then a booster in responses following additional antigen exposure (vaccination). This cannot be shown in the same graph without indicating, firstly, which subjects were analyzed after infection/vaccination, and/or at which point the vaccination occurred. Moreover, can the authors explain why there was an increase in T cell responses to the N protein considering that the N protein isn’t included in vaccine formulas. Are the authors sure there wasn’t reinfection. It seems extremely unlikely that N-specific T cell responses would continue to expand after 300 days of infection and even less likely that would expand from antigen exposure lacking this protein.

Lines 247-254: I’m not sure what this adds to the discussion. The methodology is performed via a kit that has been used by many worldwide. There is no need to justify the use of this kit and detracts from the data at hand. There’s also no need to justify using ELISpot as a technique. Again, ELISpot is a standard practice done by many worldwide.

Line 255: “Studying SARS-COV-2 vaccinated-only population cellular immunity”, this sentence doesn’t make any sense.

Lines 255-263: This paragraph is very poorly written and is incomprehensible. Please revise.

Lines 264-277: The authors write about previous studies detecting B and T cell responses up to a year after infection when this could easily be summarized into one or two sentences maximum. There is no contrast of their results to the literature until the last sentence of the paragraph. Thereafter, there is no mention of what the authors have added to the already big body of literature they have mentioned.

Lines 278-281: The authors mention a study (reference 33) that apparently showed cellular responses up to 20 months since S antigen exposure. However, while the authors of this cited paper do the same, there is no actual data to show this in the results section. These authors simply compare <6 months to >6 months. Therefore, making this assumption based on this paper is wrong.

Lines 285-291: The authors make another attempt at comparing to another study looking at vaccination after infection. However, the authors say that “a second dose of vaccine did not increase immune responses”, and yet, the paper only compares humoral responses between those that received one dose and those that received two doses, which is misleading considering this is a T cell paper. In addition, this paper uses an mRNA vaccine that is not approved and in samples sizes of 5-18 per group. I would highly suggest finding better literature material to cite.

Lines 295-296: “no valuable for further analysis information”, I’m not sure what this means?

Lines 292-300: The study limitations mentioned need to be addressed much earlier. It’s extremely important that we know upfront that the authors DO NOT KNOW the type of vaccine administered and that group A is not actually an infected-vaccinated group but could be a vaccinated-infected group (“no information was available on whether subjects of group A were infected before or after the vaccination or about the exact time of the second immunological trigger”). This makes group A extremely flawed as they haven’t mentioned this details prior.

Figures

Figure legends go below the figure.

The figure legends lack descriptive information.

Figure 1 should be represented as percentages and not absolute numbers. The percentages can also be statistically analyzed.

Figure 2 should have the stats shown considering the authors conclude that groups A and C are superior to B.

Figure 3 is completely redundant since cellular immunity is not linear, and the authors have no idea if the subject was infected/vaccinated or vaccinated/infected and at which time that was. The authors need to either separate those that got infected/vaccinated to those that got vaccinated/infected or indicate somehow at which point the subjects shown got vaccinated. Are the same subjects included in these graphs twice? If yes, this needs to be mentioned.

Round 2

Reviewer 2 Report

I thank the authors for explanations provided to my questions, and I appreciate the fact that some of my remarks have been adrressed.

Despite this, most of the main limitations highlighted in the previous round of revision haven't been overcome. Substantially, no corrections to the study design, no further testing, unsatisfactory new figures in the results sections (histograms with mean values with not even error bars are not adequate for papers of this level).

And, even worst, the missing information remain largely missing. It is not possible to publish a paper based only on one laboratory assay, with no selection criterium for subjects, no info about vaccine types or timing, and so on.

Providing a 22 lines-long paragraph of 'study limitations' is not sufficient to fix the problems, but only reminds the potetntial readers that there are too many.

I'll suggest the editor to provide an extension time for an eventual further revision, hoping it may help.

Reviewer 3 Report

Firstly, I would like to congratulate the authors on the revised manuscript. It reads so much better and it is much easier to follow. The cohort is well-described and the data is much clearer. Although there are some flaws in the study design, such as the lack of data on the type of vaccine administered, the authors have now made this clear and the data can be interpreted with this in mind.

I only have a few small issues that need to be amended:

1) On lines 17-18, "COVID-19 infection" is not a thing as COVID-19 is the disease not an infectious agent. It should be "SARS-CoV-2 infection" (i.e., the infectious agent). 
2) On line 221, there is a typo in the P value. I think there should be an "=" and not a "<". 
3) On line 263, "as regards" should be "with regards"  
4) On line 264, "have documented" should be "has documented". 

Round 3

Reviewer 2 Report

I thank the authors for the explanations, but I hope they can understand my point of view. The second major revision step was an extraordinary measure meant to provide the authors with some more time for upgrading their paper in a really substantial way. Since, as they admit, there is no way for doing it, I'm sorry I'll have to propose the rejection of the paper
